# COVID-19 and Gestational Diabetes: The Role of Nutrition and Pharmacological Intervention in Preventing Adverse Outcomes

**DOI:** 10.3390/nu14173562

**Published:** 2022-08-30

**Authors:** Ruben Ramirez Zegarra, Andrea Dall’Asta, Alberto Revelli, Tullio Ghi

**Affiliations:** 1Obstetrics and Gynaecology Unit, Department of Medicine and Surgery, University of Parma, 43126 Parma, Italy; 2Department of Obstetrics and Gynaecology, University Hospital Rechts der Isar, Technical University of Munich, 81675 Munich, Germany; 3Sant’Anna Hospital, Department of Surgical Sciences, University of Turin, 10126 Turin, Italy

**Keywords:** COVID-19, SARS-CoV-2, gestational diabetes, nutrition, physical activity, insulin, metformin, adverse maternal and neonatal outcomes

## Abstract

Pregnant women with GDM affected by COVID-19 seem to be at higher risk of adverse maternal and neonatal outcomes, especially those with overweight or obesity. Good glycemic control seems to be the most effective measure in reducing the risk of GDM and severe COVID-19. For such purposes, the Mediterranean diet, micronutrient supplementation, and physical activity are considered the first line of treatment. Failure to achieve glycemic control leads to the use of insulin, and this clinical scenario has been shown to be associated with an increased risk of adverse maternal and neonatal outcomes. In this review, we explore the current evidence pertaining to the pathogenesis of SARS-CoV-2 leading to the main complications caused by COVID-19 in patients with GDM. We also discuss the incidence of complications caused by COVID-19 in pregnant women with GDM according to their treatment.

## 1. Introduction

In December 2019, severe acute respiratory syndrome coronavirus 2 (SARS-CoV-2) was first reported in Wuhan, China [1]. Months later, the disease caused by the virus—i.e., COVID-19—was classified by the WHO as a pandemic [2]. During the course of the disease, some of the patients affected by COVID-19 may develop a severe disease and show the deterioration of their clinical conditions. This has been reported to occur approximately one week after the onset of symptoms [3]. Severe COVID-19 is a life-threatening complication commonly associated with acute respiratory distress syndrome, thromboembolic complications, and disorders of the central or peripheral nervous system, resulting in multiorgan failure and death [4]. Since the beginning of the pandemic, pregnancy has been considered an independent risk factor for severe COVID-19 [5,6,7], especially if associated with other comorbidities, such as older age, hypertension, diabetes mellitus, obesity, smoking, and cardiovascular diseases [8,9,10]. Furthermore, severe COVID-19 has also been associated with increased maternal and perinatal morbidity and mortality [6,7,10,11,12].

Gestational diabetes mellitus (GDM) is defined as glucose intolerance first identified during pregnancy and usually resolving after birth [13]. It is one of the most common medical complications of pregnancy, with a prevalence ranging from 5.8% (1.8–22.3%) in Europe to 12.9% (8.4–24.5%) in the Middle East and North Africa [14,15]. Although the etiology of GDM has yet to be fully understood, different risk factors for the development of the disease, including maternal age, obesity, increased gestational weight gain, and a family history of diabetes mellitus, have been identified [14,16]. Pregnant women with GDM are at increased risk of short-term adverse maternal and perinatal outcomes, such as cesarean section, shoulder dystocia, preeclampsia, fetal macrosomia, neonatal hypoglycemia, and the admission of neonates to the intensive care unit [14,17,18]. Additionally, mothers with GDM are at higher risk of developing type 2 diabetes mellitus, metabolic syndrome, obesity, and cardiovascular disease later in life [19,20,21,22]. There is also growing evidence for the effects of GDM and the long-term metabolic risk conferred to the offspring through a process called fetal programming [23]. According to this evidence, persons born to a mother with GDM are more likely to develop glucose intolerance, type 2 diabetes mellitus, obesity, metabolic syndrome, or cardiovascular diseases during their lifetime [24,25,26,27]. The main goal of the management of GDM is good glycemic control, which is associated with improved maternal and fetal outcomes [28,29,30]. Treatment for GDM aims to maintain serum glucose levels within normal ranges. The first-line intervention for GDM includes dietary advice and lifestyle modifications. However, some patients fail to maintain euglycemia and may require pharmacological treatment, such as insulin and/or oral metformin [31,32].

Since the beginning of the COVID-19 pandemic, there has been a growing interest in the association between GDM and COVID-19. On the one hand, SARS-CoV-2 may determine hyperglycemia and diabetes mellitus due to its interaction with the angiotensin-converting enzyme 2 (ACE2) receptors and the resulting damage to the pancreatic islet cells [33,34,35,36]. On the other hand, pregnant women with GDM are more likely to manifest symptoms of COVID-19 than their counterparts, due to the common presence of other comorbidities such as obesity and hypertension [37,38,39]. Importantly, mothers with GDM are more likely to develop severe COVID-19 [10,37,40], be admitted to the intensive care unit [41,42], and develop other adverse maternal and neonatal outcomes [37,39].

In this review, we explore the current evidence pertaining to the pathogenesis of SARS-CoV-2 leading to the main complications caused by COVID-19 in patients with GDM. We also discuss the impact of nutrition on GDM and COVID-19 and the incidence of complications caused by COVID-19 in pregnant women with GDM according to the treatment received.

## 2. Materials and Methods

We performed a literature search in the electronic databases PubMed and Medline, focusing on COVID-19 and GDM. For such purposes, a combination of the following keywords (MeSH) was used: “gestational diabetes mellitus”, “COVID-19”, “SARS-CoV-2”, and “micronutrients”. Only publications in English were considered. We also manually searched through the references of the selected publications for additional relevant articles and included them in our review. We focused mainly on meta-analysis, randomized controlled trials, and large prospective or retrospective cohort studies. As the present article is considered an expert review, we did not perform a systematic review of the literature.

## 3. Current Evidence on the Association between COVID-19 and Diabetes in Pregnancy

Since the beginning of the COVID-19 pandemic, the prevalence of GDM has increased compared to previous years [43,44,45]. Different theories regarding the pathophysiology of SARS-CoV-2 infection in relation to the occurrence of GDM have been proposed, including an increase in cases of newly diagnosed GDM or failure to comply with the first-line treatment for GDM due to worsening hyperglycemia (see next section).

The entry point of SARS-CoV-2 into human cells is the ACE2 receptor [33]. SARS-CoV-2 uses a highly glycosylated spike protein to bind to the cell surface ACE2 receptor, a cell receptor that is also glycosylated [34]. Notably, increased glycosylation in a variety of cells and tissues is frequent in patients with diabetes mellitus, which may facilitate the entry of SARS-CoV-2 into the host cell [46]. ACE2 receptors are expressed in the cells of most organs and tissues, including the pancreatic islet cells, and are overexpressed in patients with diabetes mellitus [47]. When SARS-CoV-2 binds to the ACE2 receptor, the ACE2 pathway is activated, causing acute ß-cell dysfunction and leading to a hyperglycemic state, which may increase the severity of GDM or promote the de novo onset of GDM [33,35,48]. A continuous hyperglycemic state also increases viral replication and suppresses the antiviral immune response in the pregnant tissues, such as pulmonary epithelial cells [49,50].

Currently, the pathophysiology of SARS-CoV-2 is well understood and can explain why pregnant women with GDM are more likely to acquire COVID-19 compared to healthy mothers [37,38,42] and even to mothers with other comorbidities, such as cardiac diseases, hypertension, or asthma [38]. Additionally, it also explains the increased hospitalization rates for COVID-19 observed in mothers with GDM [51]. Notably, hyperglycemia at the time of hospital admission for COVID-19 has been associated with worse prognosis in pregnant patients with diabetes [39]. Nevertheless, there is an open question regarding new-onset GDM secondary to COVID-19 and its implications during pregnancy. Data on this subject is lacking, and no conclusions can currently be made. Further research is warranted to evaluate whether the outcome of pregnancies complicated by new-onset GDM alone differs from that of pregnant women diagnosed with COVID-19 and new-onset GDM.

Severe COVID-19 is caused by an excessive and aberrant cytokine storm as a result of a rapid increase in proinflammatory cytokines, which is driven by an exaggerated host immune response [52]. One of the most important cytokines is interleukin 6 (IL-6), whose levels correlate with the severity of COVID-19 [53,54]. One of the main sources of proinflammatory cytokines is adipose tissue [55]. Increased adipose tissue promotes macrophage infiltration and the increased production of inflammatory cytokines such as leptin, tumor necrosis factor alpha (TNF-α), and IL-6 [56]. Elevated levels of IL-6 are also associated with insulin resistance, hyperglycemia [57], and obesity as well as with diabetes mellitus and GDM [56,57,58,59]. As a result, pregnant women with GDM, especially with a body mass index (BMI) > 25 kg/m^2^, are at increased risk of developing severe COVID-19 and being admitted to an intensive care unit [10,37,39,40].

In summary, data support the notion that mothers with GDM are at high risk of being hospitalized due to COVID-19 and developing a severe form of the disease, especially if obesity is also present. Good glycemic control may have beneficial effects on clinical outcomes in patients with GDM and COVID-19.

## 4. Gestational Diabetes Treated with Diet and COVID-19

Overall, 80% of women diagnosed with GDM can maintain glycemic control through dietary advice and lifestyle modifications [31,60]. This might not hold true for patients with GDM and COVID-19, as the rates of patients treated only with diet modifications have been shown to be lower (approximately 60%) [39].

Dietary modification is the first-line treatment for GDM. The Mediterranean diet, which consists mainly of vegetables, legumes, nuts, cereals, and fish, is effective in improving glycemic control and reducing the risk of GDM and associated adverse outcomes [58,61]. In detail, the Mediterranean diet has been associated with lower weight gain during pregnancy [62] and an improvement in short- [63,64,65] and long-term maternal and fetal outcomes [66,67,68,69] in patients with GDM. Conversely, the Western diet, which is characterized by a high consumption of sugars, proteins, and saturated fats, is associated with obesity, type 2 diabetes mellitus, the activation of the innate immune system, and the impairment of adaptive immunity [58]. The latter two conditions lead to chronic inflammation and impaired host defense against viruses, such as COVID-19, and can be counted among the risk factors for severe disease [70]. Containment measurements adopted during the epidemic waves of SARS-CoV-2 infection, such as lockdowns, led to an increase in the consumption of sugary food and snacks, leading to poor glycemic control in women [71,72,73] and hence potentially worsening maternal and fetal outcomes.

Another important topic to discuss when addressing nutrition in patients with GDM is nutrient deficiency. Deficiencies in vitamin D, vitamin E, zinc, and magnesium have been found in mothers with GDM and are associated with chronic low-level inflammation and oxidative stress [74,75,76,77]. Available data from one meta-analysis involving 12 randomized controlled studies found that vitamin and mineral supplementation (vitamin D, vitamin E, magnesium, zinc, calcium, and selenium) improved glycemic control and attenuated low-grade chronic inflammation and oxidative stress in mothers with GDM [78]. Myo-inositol, a sugar found in grains, corn, nuts, meat, legumes, and fresh citrus fruits, has also been associated with a reduction in the incidence of GDM and fetal macrosomia in normal and overweight/obese mothers [79,80,81]. Myo-inositol reduces serum glucose and improves insulin sensitivity [82] in a similar way to metformin [83]. Recently, a lot of emphasis has been placed on the use of probiotics for the prevention of GDM [84]. Probiotics have the ability to modify the intestinal microflora, increasing the degradation of polysaccharides [85], and secrete proinflammatory mediators, reducing local and systemic inflammation [86]. A randomized controlled trial [85] and a subsequent meta-analysis [87] reported the reduced frequency of GDM in patients receiving probiotics. Although data regarding micronutrients/probiotics and COVID-19 are lacking, deficiencies in the aforementioned micronutrients might be associated with poor glycemic control and low-grade chronic inflammation, both of which might facilitate progression to severe disease [49,50,53,54]. According to the available evidence, it is important to offer dietary advice to all pregnant patients, especially those with GDM and a high BMI. Overall, a good dietary approach involving a Mediterranean diet and micronutrient/probiotic supplementation might have the potential to reduce the risk of GDM, SARS-CoV-2 infection, and severe COVID-19. More research is needed in this area to determine the real benefit of micronutrient and probiotic supplementation in patients with GDM and COVID-19. A summary of the most important dietary interventions is shown in Figure 1.

Lifestyle modifications including physical activity are also among the first-line options for tackling GDM and poor glycemic control in women with GDM. In women with GDM, physical activity has been shown to improve glucose control and reduce insulin use [88,89], especially when combined with dietary modifications [90]. As mentioned above, one of the measures adopted during the COVID-19 pandemic that had a negative impact on the prevalence of GDM was lockdowns [43]. Lockdown periods led to unhealthy diets and reduced physical activity in some individuals [91,92], resulting in a rise in insulin resistance, total body fat, abdominal fat, and inflammatory cytokines [93]. Moreover, mothers suffered from psychological stress, depression, and anxiety during quarantine, which contributed further to the increase in unhealthy diets and the reduction in physical activity, worsening the rates of hyperglycemia [92]. These COVID-19-related containment measurements led to increased HbA1c concentrations [72] and poor glycemic control [73,94], which might also explain the increasing rates of insulin use among mothers with GDM during the SARS-CoV2 pandemic.

Mothers with a normal BMI and GDM on diet treatment are not at higher risk of contracting or developing symptomatic COVID-19 compared to women with pregnancies that are not complicated by GDM [37]. Conversely, overweight or obese mothers with GDM on diet treatment have a 35% higher risk of developing a symptomatic disease [37]. However, this is not reflected in the maternal or neonatal outcomes, as mothers with GDM on diet treatment maintain good glycemic control. This might explain the low rates of maternal and neonatal complications, especially in patients with a BMI < 25 kg/m^2^. However, it is important to underline that the available evidence is limited, and the studies so far published may not have been capable of detecting significant changes in adverse maternal and neonatal outcomes. Therefore, results should be interpreted with caution, especially when counseling and treating a pregnant patient with overweight or obesity and GDM, as obesity alone is an acknowledged risk factor for severe COVID-19 [42,95].

To date, evidence regarding adverse maternal and neonatal complications in patients with GDM treated with diet and lifestyle modifications is still lacking. Research in this area should be encouraged in order to better understand the prognosis of these patients (especially those with a higher BMI) when affected by COVID-19.

## 5. Gestational Diabetes Treated with Insulin Therapy and COVID-19

The use of insulin in patients with GDM is recommended when glycemic control is not achieved within two weeks of diet and lifestyle intervention treatment [31,32]. Good glycemic control has been shown to reduce maternal and perinatal morbidity in pregnancies complicated by GDM and COVID-19 [33,34,35,48,49,50]. However, since the beginning of the COVID-19 pandemic in 2020, there has been a 30% increase in the use of insulin to control glycemia in patients with GDM compared to previous years [39,94]. This is likely due to the effect of lockdowns on physical activity and diet, resulting in an increase in serum glucose levels [91,92].

Evidence on the occurrence of complications associated with COVID-19 in pregnant patients with GDM treated with insulin is limited. Similarly to non-pregnant individuals with diabetes mellitus treated with insulin [96,97], mothers affected by GDM who are on insulin are also more likely to have a confirmed SARS-CoV-2 infection irrespective of their BMI [37].

Patients affected by COVID-19 and diabetes mellitus type 2 treated with insulin are at increased risk of developing severe/critical complications or dying [98]. Similarly, mothers with GDM on insulin treatment may undergo a more severe course of the disease and develop adverse maternal outcomes, especially if they are overweight or obese [39]. The mechanisms underlying this association are yet to be fully understood. One hypothesis is that insulin may increase the levels of proinflammatory cytokine from activated macrophages and promote a proinflammatory state, which may exacerbate the inflammation cascade caused by COVID-19 [99]. Another hypothesis is that insulin may increase the susceptibility to lung inflammation [100] and hence may worsen the pulmonary complications associated with COVID-19. Regarding neonatal outcomes, data from the German COVID-19 registry (CRONOS) showed an almost five-fold increased risk of adverse neonatal outcomes in mothers with GDM treated with insulin with a normal BMI. On the contrary, treatment with insulin in overweight or obese mothers with GDM has not been shown to be independently associated with adverse neonatal outcomes. It is important to note, however, that the incidence of adverse neonatal outcomes in this group was two-times higher than in mothers with no GDM and a normal BMI (25.0% vs. 12.3%); therefore, the limited number of cases so far reported may have led to a false negative result.

Most of the data on the outcomes of COVID-19 patients with pregnancies complicated by diabetes and taking insulin is lacking. Only one study has compared maternal and neonatal outcomes in mothers with GDM and insulin treatment. Therefore, caution is needed before reaching conclusions regarding the progression of the disease in this population. Patients with GDM and insulin treatment should be considered high-risk patients, especially when characterized by a high BMI, and should be monitored closely, as this population has been shown to be affected more often by severe COVID-19. Moreover, vaccination and protective measurements against COVID-19 should be recommended, especially to this subgroup of patients.

## 6. Gestational Diabetes Treated with Metformin and COVID-19

The use of metformin in patients with GDM is recommended when they fail to achieve glycemic goals with diet and lifestyle interventions [101,102,103]. These recommendations are based on randomized clinical trials [104] and meta-analyses [105,106], which have shown similar outcomes when compared to patients with GDM treated with insulin.

There is no data regarding maternal or perinatal outcomes in patients with COVID-19 and GDM treated with metformin. Available information on the outcomes of COVID-19 in patients treated with metformin pertains to patients with type 2 diabetes mellitus. As above, several theories have been proposed to explain the association between insulin and adverse outcomes [99,100]. Moreover, the key molecular target of metformin is 5’-AMP-activated protein kinase, which mediates the expression and stability of ACE2 receptors [107,108]. The underexpression or instability of ACE2 receptors may impair the entry of SARS-CoV-2 into the host cells. Another hypothesis is related to the reduction in the levels of TNF-α that has been associated with the administration of metformin [109,110], which suggests that the drug has anti-inflammatory properties. Therefore, it is plausible that pregnancies treated with metformin instead of insulin are at a lower risk of adverse outcomes and COVID-19-related mortality [111,112,113,114]. Further studies are needed to clarify the interaction between metformin and the expression of ACE2 receptors, which is key for the entrance of SARS-CoV-2 into the host cells. Notably, current recommendations state that metformin treatment should be stopped in patients with respiratory distress, renal impairment, or heart failure, due to the increased risk of lactic acidosis [115,116].

## 7. Future Perspectives

In the present expert review, we showed that dietary and lifestyle modifications during pregnancy provide a window of opportunity for healthcare professionals to reduce the risk of developing GDM. However, there is no evidence as to how these interventions might impact the course of COVID-19 infection and its related outcomes in patients with GDM.

Substantial evidence regarding maternal and neonatal outcomes in pregnant patients affected by COVID-19 and GDM, especially according to the type of treatment, is lacking. This information is critical for counseling patients affected by these two conditions. Understanding whether metformin could be a potential alternative to insulin in mothers with GDM, especially with a higher BMI, is a subject of further research given that insulin use has been associated with a progression to severe disease.

## 8. Conclusions

In summary, the available data suggest that pregnant women with GDM are at increased risk of adverse outcomes when infected with SARS-CoV-2. Dietary modifications seem to be the easiest and most reliable way to maintain glycemic control and improve immune function, hence decreasing the risk of COVID-19 and its sequelae in women with GDM. Moreover, physical activity might further improve glycemic control and reduce insulin resistance. These approaches have the potential to reduce the severity of the adverse outcomes caused by COVID-19. Mothers with GDM under insulin treatment are at increased risk of adverse outcomes, especially in cases of overweight or obesity. Therefore, pregnant patients with diabetes mellitus or obesity should be regarded as a high-risk population susceptible to severe COVID-19 disease. The use of insulin appears to be associated with negative effects in patients with GDM, but this seems to be related to the deeper impairment of the metabolic control of such patients and not to the use of the insulin per se. Metformin might be used as an alternative to insulin in patients with COVID-19; however, more studies are needed before this drug treatment can replace insulin therapy.

## Figures and Tables

**Figure 1 nutrients-14-03562-f001:**
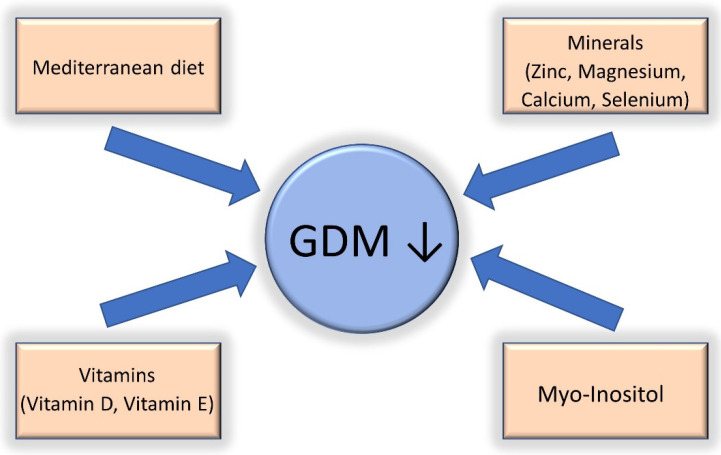
Nutritional interventions that have been shown to be effective in the reduction of gestational diabetes (GDM).

## Data Availability

Not applicable.

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
