# Peer review of "COVID-19 and Gestational Diabetes: The Role of Nutrition and Pharmacological Intervention in Preventing Adverse Outcomes"

_nutrients, 2022, doi:10.3390/nu14173562_

Round 1

Reviewer 1 Report

This review provides nice overview of covid-19 and GDM. There are few concerns: 

As the topic suggests, it would be interesting to read if prenatal vitamin deficiency or other nutrients during early pregnancy is associated with GDM and later consequences in the offspring, and how covid-19 might contribute to these associations.

Not much evidence on neonatal/fetal outcomes and complications presented to support the conclusion made "In summary, available evidence suggests that pregnant women affected by GDM or obesity are at increased risk of adverse fetal outcomes, severe complications and maternal death, when COVID-19 infection supervenes." Perhaps a separate heading for the topic would interest readers.

Minor comment:

Section 4. last two paragraphs. Clarify if both proposed explanation are based on rodents or human studies? Also a reference after second point (sentence) is missing. 

Author Response

REVIEWER 1

This review provides nice overview of covid-19 and GDM. There are few concerns: 

  1. As the topic suggests, it would be interesting to read if prenatal vitamin deficiency or other nutrients during early pregnancy is associated with GDM and later consequences in the offspring, and how covid-19 might contribute to these associations.

Reply: We thank the reviewer for this comment. As requested, we have added an entire paragraph on the section “Gestational diabetes treated with diet and Covid-19” focusing on vitamin, mineral, Myo-Inositol and probiotics supplementation during pregnancy, and how they might impact the outcomes of COVID-19.

  1. Not much evidence on neonatal/fetal outcomes and complications presented to support the conclusion made "In summary, available evidence suggests that pregnant women affected by GDM or obesity are at increased risk of adverse fetal outcomes, severe complications and maternal death, when COVID-19 infection supervenes." Perhaps a separate heading for the topic would interest readers.

Reply: We thank the reviewer for the comment. We have completed the current review with new data reporting on adverse maternal and neonatal outcomes in the different treatment group of patients with GDM. We have therefore not added a separate heading for this topic, and we have included the information in the corresponding sections.

Minor comment:

  1. Section 4. last two paragraphs. Clarify if both proposed explanation are based on rodents or human studies? Also a reference after second point (sentence) is missing. 

Reply: We thank the reviewer for the comment. The first explanation are based on human studies, thus we did not feel the necessity to clarify it, as we did in point 2). We have added the missing reference for the point 2).

Reviewer 2 Report

I think that the title of the article is not appropriate. Nutrition concerns only one chapter of the whole manuscript. Keywords used include: COVID19 and Gestational Diabetes; GDM; COVID19; pregnancy; angiotensin—converting enzyme 2 receptors. No word such as nutrition or diet is included.

The abstract section should be revised completely so as to be a short version of the whole manuscript. At its current form it looks like an introduction with a closing aim sentence.

Also most of the manuscript seems to be a referral to the INTERCOVID study. If this is the case, what is the additional data that the manuscript offers?

In the last chapter the authors note "there is no data regarding maternal or perinatal outcomes in patients with COVID19 and GDM treated with metformin", so this chapter should be removed.

The authors also note "there is no study focusing on the incidence of maternal and neonatal complications in patients with GDM treated with diabetic diet" & later on "evidence on the occurrence of complications associated with COVID-19 in pregnant patients with GDM is limited". That being true the whole review fails. At least there should be a limitations section emphasizing the lack of data otherwise the manuscript should focus on DM and COVID-19 instead of GDM and COVID-19.

I also have some concerns for the citations used. As an example, the authors note "Maintaining a good glycemic control has been found to be a good measure to reduce morbidity in mothers and fetuses with GDM and COVID-19[25]." Ref 25 dates back in 2010. There was no COVID-19 in 2010!

It is mandatory that a methods section is added.

Author Response

REVIEWER 2

  1. I think that the title of the article is not appropriate. Nutrition concerns only one chapter of the whole manuscript. Keywords used include: COVID19 and Gestational Diabetes; GDM; COVID19; pregnancy; angiotensin—converting enzyme 2 receptors. No word such as nutrition or diet is included.

Reply: We thank the reviewer for the comment. We have changed the title, as it now reads “COVID-19 and Gestational Diabetes: The Role of Nutrition and Pharmacological Intervention in Preventing Adverse Outcomes”. Moreover, we have replaced the keyword “angiotensin—converting enzyme 2 receptors” for “nutrition”.

  1. The abstract section should be revisedcompletely so as to be a short version of the whole manuscript. At its current form it looks like an introduction with a closing aim sentence.

Reply: We thank the reviewer for the comment. We have revised and amended the abstract, as requested.

  1. Also most of the manuscript seems to be a referralto the INTERCOVID study. If this is the case, what is the additional data that the manuscript offers?

Reply: We thank the reviewer for the comment, but we respectfully disagree with him. When we drafted the manuscript, the INTERCOVID study was the biggest study on GDM and COVID-19, and therefore had been cited throughout the entire manuscript. However, since then, some new manuscripts have appeared on the subject and had been added to the review.

  1. In the last chapter the authors note "there is no data regarding maternal or perinatal outcomes in patients with COVID19 and GDM treated with metformin", so this chapter should be removed.

Reply: We thank the reviewer for the comment, but we respectfully disagree with him. We believe that this section is necessary, regardless the unavailability of data on COVID-19 and GDM. As explained on the manuscript, metformin might be used as an alternative to insulin, and is associated with better outcomes. The purpose of this paragraph, is to make a call to all people interested in this topic in order to increase the number of publications on this subject.

No change was made to the manuscript.

  1. The authors also note "there is no study focusing on the incidence of maternal and neonatal complications in patients with GDM treated with diabetic diet" & later on "evidence on the occurrence of complications associated with COVID-19 in pregnant patients with GDM is limited". That being true the whole review fails. At least there should be a limitations section emphasizing the lack of data otherwise the manuscript should focus on DM and COVID-19 instead of GDM and COVID-19.

Reply: We thank the reviewer for the comment. As mentioned above, new evidence have aroused and has been included on the manuscript. We have also added a new section called “Future perspectives” where we address the limitations of the manuscript and describe what is needed to be done in order to overcome them.

  1. I also have someconcerns for the citations used. As an example, the authors note "Maintaining a good glycemic control has been found to be a good measure to reduce morbidity in mothers and fetuses with GDM and COVID-19[25]." Ref 25 dates back in 2010. There was no COVID-19 in 2010!

Reply: We thank the reviewer for the comment. The citation was incorrectly placed there and replaced with the corresponding citations.

  1. It is mandatory that a methods section is added.

Reply: We thank the reviewer for the comment. As requested, a materials and methods section had been added.

Editor

  1. We found the repetition rate of the manuscript is slightly higher, please modify the duplication when revising the manuscript. You could find the duplication report in attach.

Reply: We thank the editor for the comment. We have revised and modified extensively the manuscript.

  1. We sincerely encourage you to add more content during revision, since it is our proposal that almost 4000 words in the main text.

Reply: As requested, we have added more content to our manuscript (3277 words). We have also added a figure to the manuscript.

Round 2

Reviewer 1 Report

The authors have addressed previous concerns. However, they have added new paragraphs and texts to the manuscript and it would have been easier to follow if the authors had mentioned in detail where the correction was made in the response file. 

I only have some comments on the current version. 

Materials and methods: Could authors add information on the detail of literature search made? Until when was the search made (date) and how many studies were included or excluded?

Section 4, page 5: Correction of word "pandemy" to pandemic?

Reviewer 2 Report

Authors have addressed most issues raised at the review process and significant changes (improvements) have taken place. I support the publication of the manuscript.